# Towards Global Optimal Visual In-Context Learning Prompt Selection

**Chengming Xu**[1][*]   **Chen Liu**[2][*]   **Yikai Wang**[1][✉]   **Yuan Yao**[2]   **Yanwei Fu**[1]

[1]Fudan University    [2]Hong Kong University of Science and Technology

{cmxu18, yanweifu}@fdu.edu.cn   cliudh@connect.ust.hk
yi-kai.wang@outlook.com   yuany@ust.hk

## Abstract

Visual In-Context Learning (VICL) is a prevailing way to transfer visual foundation models to new tasks by leveraging contextual information contained in in-context examples to enhance learning and prediction of query samples. The fundamental problem in VICL is how to select the best prompt to activate its power as much as possible, which is equivalent to the ranking problem of testing the in-context behavior of each candidate in the alternative set and selecting the best one. To utilize a more appropriate ranking metric and more comprehensive information among the alternative set, we propose a novel in-context example selection framework to approximately identify the global optimal prompt, i.e. choosing the best performing in-context examples from all alternatives for each query sample. Our method, dubbed Partial2Global, adopts a transformer-based list-wise ranker to provide a more comprehensive comparison within several alternatives and a consistency-aware ranking aggregator to generate globally consistent ranking. The effectiveness of Partial2Global is validated through experiments on foreground segmentation, single object detection and image colorization, demonstrating that Partial2Global selects consistently better in-context examples compared with other methods, and thus establishes the new state-of-the-arts. Code is available at https://github.com/chmxu/ranking_in_context.git.

## 1   Introduction

Foundation models like those used in AIGC, such as GPT and Gemini, play a big role in its success. Different ways to fine-tune these models are being suggested to use them more effectively. One popular idea is called Visual In-Context Learning (VICL), which is a way to teach Visual Foundation Models (VFMs) using context from images. This helps the models learn better and make more accurate 'guesses' based on what they see. Basically, VICL gives the models prompt examples to learn from, like pairs of images and labels, termed *in-context examples*, based on which *query* samples are inferred. This mimics how humans learn with the guidance. Using this method, VFMs can take lots of different tasks, like understanding point clouds [6], editing images [29] , as well as multi-modal inference [34], even if they were not trained specifically for those tasks.

The main challenge of VICL is picking the best in-context example that matches the query images, so the VFM can perform well. Empirically, we have found that randomly picking an example does not always work well, while a carefully selected example can really boost performance, as observed in

---

[*]These authors contributed equally to this work. [✉]Corresponding author.

Dr. Fu is with School of Data Science in Fudan, Fudan ISTBI—ZJNU Algorithm Centre for Brain-inspired Intelligence, Shanghai Key Lab of Intelligent Information Processing, and Technology Innovation Center of Calligraphy and Painting Digital Generation, Ministry of Culture and Tourism, China. Dr. Xu is now with Tencent Youtulab.

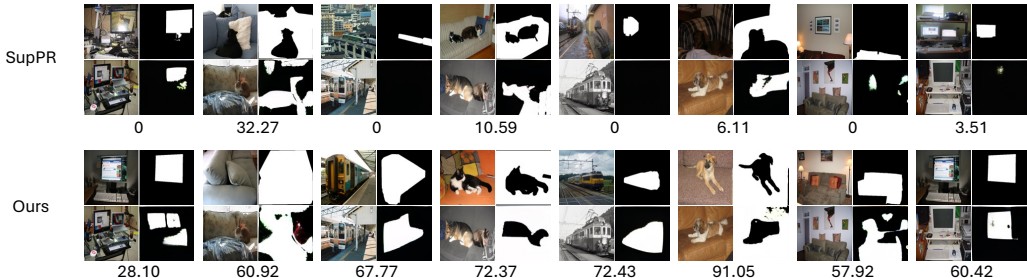

Figure 1: Qualitative comparison between our method and VPR, specifically SupPR, in foreground segmentation. In each item we present the image grid in the same order as the input of MAE-VQGAN, i.e. in-context example and its label in the first row, query image and its prediction in the second row. The IoU is listed below each image grid.

previous works [32, 21], and visualized comparison in Fig. 1. Typically, VICL has a set of options to choose from (like the training data). We can think of picking the best in-context example as a ranking problem – we want to choose the one that helps the most. Ideally, we would like to test each option with a new image, rank how well they work, and then pick the best one. However, doing that for every image is not practical. So, one of the biggest challenges in VICL is figuring out how to do this ranking without actually testing each option.

There are a few key challenges in choosing the best prompt for VICL. **(1)** *Choosing the Right Metric*: Since we cannot directly test how well a prompt works on a specific image, we need to find a different way to measure it. Previous methods like VPR [32] mainly use two metrics from ranked observations in the training set: visual similarity and pair-wise relevance instantiated through contrastive learning. However, visual similarity does not always work well as in our experiments. The contrastive learning method naturally restricts how much the model can learn because of how its objective function is designed. To make contrastive learning effective, VPR needs to collect the most representative positive and negative examples for each query, which are the best and worst in-context examples. Consequently, a large amount of ranking observations remain unused during training. *(2) Deciding on the Comparison Set*: When considering the ranking metric, it is best to exhaustively compare each example against all the candidates available. However, directly ranking all alternatives globally is often not always possible. Instead, we have to aggregate rankings from partial predictions. Pair-wise ranking, like in VPR, ensures consistent ranking predictions by comparing each option's similarity to the query individually. But it cannot grasp the overall relationship between options. Conversely, methods with larger capacity, such as list-wise ranking, struggle to create a unified ranking due to inconsistencies among different partial predictions. Anyway, balancing feasibility and capacity is crucial for VICL sample selection algorithms.

To solve these problems, this paper proposes a systematic VICL example selection framework, dubbed Partial2Global, towards the global optimal in-context examples selection. Partial2Global involves several *list-wise partial rankers* based on transformers to rank multiple alternative samples, and a *consistency-aware ranking aggregator* to achieve globally consistent ranking from the partial rankers. (1) To approximate the ground-truth ranking metric, we propose a transformer-based list-wise ranker to rank multiple alternative samples in a more informative and comprehensive manner. This model takes features of several alternatives together with the query sample from pretrained foundation models as input. The ranking predictions can then be inferred with more comprehensive knowledge from multiple alternatives and query sample by merging information among class tokens regarding each sample. Through meta-learning based training strategy, our model can learn a generalizable ranking metric for each specific task, thus better facilitating VICL example selection. (2) To approximate the global ranking, we propose a consistency-aware ranking aggregator to strike the balance between feasibility and ranking capacity. Particularly, partial and noisy ranking predictions from the trained ranking model are collected and divided into several groups, which can be reorganized and aggregated by learning a global ranker that has minimized average distance with all ranking groups. The resulted global ranking enjoys better ranking consistency and thus provides better in-context examples to boost VICL performance.

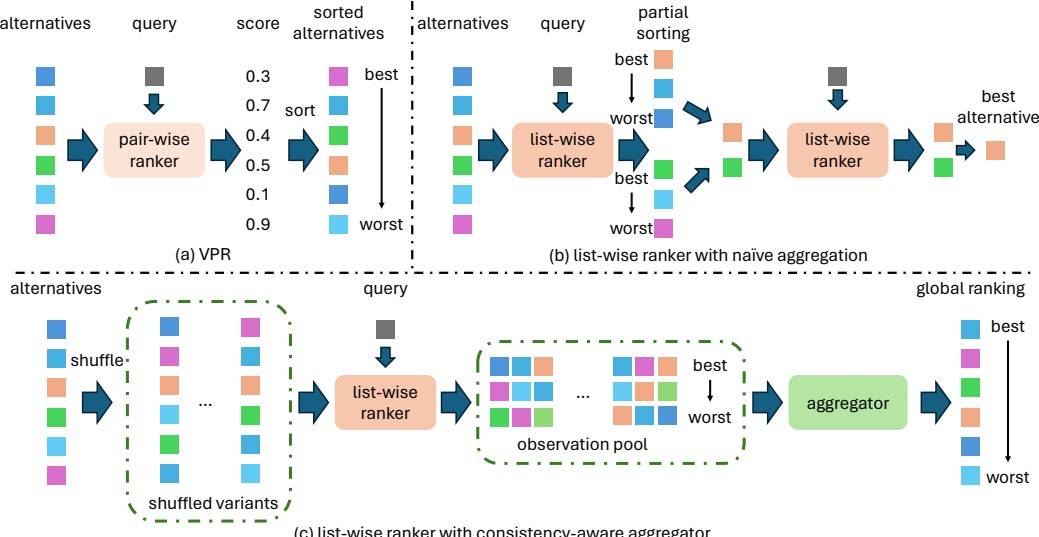

Figure 2: Systematic comparison between three different frameworks for in-context example selection. (a) VPR, which uses pair-wise ranker trained with contrastive learning to calculate the relevance score for each alternative. (b) List-wise ranker with naive aggregation, in which alternatives are split into non-overlapped subsets. These subsets are ranked with the proposed list-wise ranker. Then we iteratively select the best example in each subset and rank them. (c) List-wise ranker with consistency-aware aggregator, in which alternatives are first shuffled and predicted with list-wise ranker into an observation pool. These partial observations are then aggregated with the proposed aggregator to achieve a global ranking.

In order to validate the effectiveness of our method, we conduct experiments on three different tasks including foreground segmentation, single object detection and image colorization with datastet including Pascal VOC and ImageNet, following the previous works. Extensive results show that Partial2Global not only can provide consistently better in-context examples, which results in new state-of-the-art performance, but also shed light on the selection heuristics of in-context examples, facilitating further research on visual in-context learning.

**Contributions** of this work are as follows:

- We emphasize the importance of the global consistent ranking relationships and correct ranking metric approximation in the selection of in-context samples for VICL.
- We propose to approximate the optimal global ranking with a transformer based list-wise ranker, which can process more comprehensive information and produce stronger predictions.
- We adopt a consistency-aware ranking aggregator to aggregate the partial predictions generated by our ranking model.
- Extensive experiment results show that our method consistently works well in various visual tasks based on pretrained in-context learning model, showing the practical value of such a method.

## 2 Related works

**In-context learning.** Given the current trend of scaling up model sizes, the large-scale models such as large language models (LLMs) [2] and their multi-modal counterparts [10] are shown to gradually learn the ability to perform in-context learning, i.e. inference with knowledge provided by few-shot context samples, rather than training the model with extra data. For example, Pan et. al. [15] designed an in-context learning system to generate symbolic representation, which can then be used for logical inference. Zhang et. al. [33] proposed to leverage in-context learning to edit factual knowledge in LLMs. Handel et. al. [7] showed that in-context learning works by compressing training set into task representations which can modulate models during inference for desired output. Such a property enables nonprofessional users to drive the model simply with several examples, the same as

someone teachers his children. Recently, MAE-VQGAN [1] and Painter [24] show that by learning to inpaint the image grid composited by support and query pairs, the vision models can also learn the in-context learning ability. VPR [32] follows MAE-VQGAN and focuses on the specific problem of selecting good in-context examples. They propose a simple strategy by learning a contrastive learning based metric network with performance metrics. Prompt-SelF [21] analyzes VPR's strategy and extends it to take into account both pixel-level visual similarity and patch-level similarity. Moreover, prompt-SelF proposes an test-time ensemble strategy to gather predictions with different permutation of in-context image grids. Our work inherits the idea of VPR to concentrate on in-context example selection. Different from VPR, we stress the importance of make full use of training sample, and the post-process of model predictions. To this end, we propose a novel framework which first trains ranking models with the performance metric data and then process the ranking predictions with consistency-aware ranking aggregator.

**Sample selection and Ranking in deep learning.** Sample selection is crucial for improving model training [20, 13, 27, 28, 26, 25]. In in-context learning, sample selection focuses on finding the best prompt example for a given query. This is typically done by training additional ranking models. These models compare a list of examples with the query based on their similarity in a specific property. Burges et al. [3] proposed an objective function for pairwise ranking, which was further extended to listwise ranking [23]. Recently several works focus on learning ranking with transformer structures. For example, Kumar et al. [9] proposed to adopt attention mechanism to interact between samples to be ranked. Liu et al. [11] proposed to build the ranking model based on GPT model. In this paper we propose to leverage the learning to rank technique to build a transformer-based list-wise ranker, which can learn more comprehensive information than previously used pair-wise ranker learned with contrastive learning.

**Aggregating partial ranking predictions.** Jiang et. al. [8] proposed HodgeRank as a Hodge-theoretic approach to statistical ranking, which can provide global ranking with incomplete and inconsistent ranking data. The comprehensive insight of HodgeRank made it widely applicable to quality of experience assessment. For example, Xu et al. [30] adopts HodgeRank for subjective video quality assessment based on random graph models. The similar idea is also applied to crowdsourced pairwise ranking aggregation [31]. In this paper we take inspiration from Hodge theory to build a novel framework for visual in-context learning. While such a strategy has never been studied before, we show that our framework can significantly boost the visual in-context performance by selecting better examples.

## 3 Methodology

**Preliminary: Visual in-context learning.** Visual In-Context Learning (VICL) aims to build the inference procedure of each testing sample based on knowledge provided by in-context examples. MAE-VQGAN [1] instantiates such an idea through image inpainting. Concretely, given a query sample, an example image is selected randomly along with its annotations (e.g., bounding boxes, masks, etc.) from training set. Then the example image, example label, query image and a random query label to be predicted are arranged as an image grid so that the query label is placed in the lower right corner. Then this composed image is used as input to an inpainting model. The masked patches are predicted by a transformer autoencoder and further decoded with the pretrained VQ-VAE [22]. To ensure the generalization ability, MAE-VQGAN is pretrained on a large-scale dataset collected from arxiv papers. While it shows promising results among different kinds of tasks, the important problem of in-context example selection is omitted, which leaves this method great potential for improvement.

**Preliminary: Visual Prompt Retrieval.** Visual Prompt Retrieval (VPR) [32] mainly focuses on the in-context example selection based on MAE-VQGAN. Specifically, VPR starts with the heuristic that more similar an image is to the query image, the better it is as the in-context example. Based on this, the unsupervised and supervised variants of VPR are designed. In the unsupervised setting, VPR directly selects the most visually similar training images for each query image. As for the supervised setting, VPR first creates of a performance metric dataset collected from the training set, symbolized as $\{x_q^i, \mathcal{X}_R^i, y^i\}_{i=1}^{N_{Tr}}$, where $x_q$ denotes query sample, $\mathcal{X}_R$ denotes a set of $K$ alternatives to be ranked, and $y$ denotes the performance metric, e.g. IoU for segmentation or accuracy for classification, for pretrained MAE-VQGAN when testing $x_q$ on specific task using $\mathcal{X}_R$ as in-context examples respectively. Then a metric learning model is trained to learn the in-context performance of training samples, i.e. the best performing samples are labeled as positive samples and learned

against the worst performing ones as negative samples. Such a method, while outperforming the naive MAE-VQGAN, still cannot fully explore the knowledge regarding the ranking of in-context examples with the unprocessed partial observations. To this end, we propose a novel method to boost VICL with more proper in-context example selection.

**Overview.** To boost VICL with better in-context example selection, we in this paper propose a novel pipeline including two main steps. First we train a ranking model (Sec. 3.1) based on the performance metric dataset collected in the same way as VPR. Different from VPR, we utilize an extra transformer model to learn list level ranking instead of instance level scoring so that inner relationship between alternatives can be better utilized for ranking. Once the ranking model is trained, the global ranking of alternative set needs to be gathered from partial ranking predictions. To this end, we propose to adopt a consistency-aware ranking aggregator (Sec. 3.2) to process predictions to provide globally consistent ranking results, thus better facilitating VICL.

## 3.1 Ranking in-context examples with list-wise ranker

In order to learn a more proper ranking metric approximation, we propose to adopt a transformer-based list-wise ranker. Our methodology is rooted in the principles of VPR. Given the query sample $x_q$ and the alternative set $\mathcal{X}_R$, we first sample a $k$ size subset $x$ from $\mathcal{X}_R$. Then a ranking model $\phi_k$ is built to provide ranking prediction of $x$ as $\phi_k(x, x_q)$. Concretely, features $z_q \in \mathbb{R}^{(N+1)\times C}, z \in \mathbb{R}^{k \times (N+1) \times C}$ corresponding to $x_q, x$ are extracted with pretrained transformer model $\phi$ such as CLIP [17] or DINO [4], where $N + 1$ features include class token and patch tokens, $C$ denotes feature channels. Following the feature extraction, we concatenate all these features to form a feature sequence $\hat{z} \in \mathbb{R}^{(k+1)(N+1)\times C}$. For the sake of simplicity and to facilitate further processing, we rearrange this sequence such that all class tokens are positioned at the beginning. This sequence $\hat{z}$ is then processed through several newly introduced transformer layers. These layers are designed to enhance interaction between the global-level and local-level features contained in different images. This interaction is crucial as it allows the model to gather the necessary information for ranking the alternatives. Upon completion of these layers, we collect the class tokens of alternatives $z_{cls} = \hat{z}_{[1:k+1,:]}$. These tokens are then processed through linear layers to generate the ranking prediction $\hat{y} \in \mathbb{R}^k$. This prediction $\hat{y}$ serves as an indicator of the ranking of the alternatives, providing a quantitative measure of their relevance to the query sample.

**Training objectives.** Our ranking model is optimized with a composed objective as follow:

$$\mathcal{L} = \underbrace{\sum_{i,j=1,i\neq j}^{k} \max(0, \mathbf{1}(y_i > y_j)(x_i - x_j + \delta))}_{\mathcal{L}_{margin}} + \underbrace{\text{NeuralNDCG}(\tau)(\hat{y}, y)}_{\mathcal{L}_{sort}} + \underbrace{\text{MSE}(\hat{y}', y)}_{\mathcal{L}_{reg}}, \quad (1)$$

where $\mathcal{L}_{margin}$ denotes the pair-wise marginal ranking loss, $\mathcal{L}_{sort}$ denotes list-wise Neural-NDCG [16], $\delta$ and $\tau$ represents the loss coefficients for two loss terms respectively. $\mathcal{L}_{margin}$ and $\mathcal{L}_{reg}$ act a similar role as the supervised contrastive loss in VPR, which encourages the model the predict higher score for better samples in each pair. $\mathcal{L}_{sort}$, on the other hand, can drive the model to learn inner relationship among more alternatives, thus leading to better ranking predictions.

## 3.2 Consistency-aware ranking aggregator

For this section, we will introduce how to obtain consistent global rank for the alternative set given the trained ranking model as introduced above.

**Motivation.** In the process of selecting in-context examples, it is crucial to identify the most suitable candidates from the available pool. However, the ranking model we employ only provides partial ranking predictions on a subset instead of the full alternative set. A naive solution would be splitting the alternative set into non-overlapping subsets, which are then processed with the trained ranking model respectively. Then the top ranked samples from each subset are gathered together as a new alternative set, which is then repeatedly split and ranked again until there is only one sample.

While such a method can provide reasonable results as we will show in the experiment, it can suffer from three main problems: (1) In total, $\binom{K}{k}$ subsets with size $k$ can be randomly sampled from an alternative set with size $K$. However, the naive method only utilizes $\lceil \frac{K}{k} \rceil$ subsets, which is extremely

---

**Algorithm 1** Consistency-aware ranking aggregator

---

**Input:** Train set $\mathcal{X}_{train}$, query sample $x_q$, trained ranking models $\{\phi_k\}$, alternative set size $K$.
1: Alternative set $\mathcal{X}_R = topK_{\hat{x} \in \mathcal{X}_{train}}(sim(\hat{x}, x_q))$
2: Initial preference matrix set $\mathcal{S} := \emptyset$
3: **for** rank-k model $\phi_k$ **do**
4:     Build observation pool $\mathcal{X}_k$ from $\mathcal{X}_R$
5:     **for** randomly shuffled $\mathcal{X}_k^i$ from $\mathcal{X}_k$ **do**
6:         $\mathcal{R}_k^i = \bigcup_{x \in \mathcal{X}_k^i} \phi_k(x, x_q)$
7:         Aggregate $\mathcal{S}^i$ from $\mathcal{R}_k^i$
8:         $\mathcal{S} = \mathcal{S} \bigcup \mathcal{S}^i$
9:     **end for**
10: **end for**
11: Aggregate global ranking $r$ as Eq. 3
12: **return** Top ranked sample.

---

limited and can result in selecting poor performing in-context examples. (2) Since this naive method is iterative, the selection error would accumulate, leading to unbearably wrong prediction. (3) It is inefficient to fetch other top ranked samples such as second or third best ones with this method. Therefore a new method, which can directly provide global ranking prediction by fully utilizing information contained in the alternative set, can thus better serve the purpose of in-context example selection. To this end we propose the consistency-aware ranking aggregator, which allows us to derive a global ranking from the partial information, thereby making it possible to rank all candidates in relation to each other while guaranteeing global consistency.

**Algorithm.** Our method starts with gathering enough sufficient information from all alternatives. Given the query sample $x_q$, alternative set $\mathcal{X}_R$ and a trained $k$-length ranking model $\phi_k$, we can first build an observation pool $\mathcal{X}_k = \{\mathcal{X}_R^i\}_{i=1}^{N_p}$, where $\mathcal{X}_R^i$ denotes a randomly shuffled variant of $\mathcal{X}_R$. Then for each $\mathcal{X}_R^i$ we follow the naive ranking method to split it into $\lceil \frac{K}{k} \rceil$ non-overlapped $k$-length sequences and rank these sequences with $\phi_k$, resulting in a predicting set $\mathcal{R}_k^i$. The predictions are further aggregated into a preference matrix $S^i \in \mathbb{R}^{K \times K}$. If the $m$-th alternative is favored over the $n$-th alternative, we let $S_{mn}^i = 1$ otherwise $S_{mn}^i = -1$. If no predictions are related with $m$ and $n$-th alternatives, then $S_{mn}^i = 0$. $S^i$ can then be transformed into an pair-wise indication set $E^i = \{(m,n) | S_{mn}^i \neq 0\}$. In this way we can finally get a preference matrix set $\mathcal{S} = \{S^i\}$.

Denoting the global ranking of $\mathcal{X}_R$ as a score vector $r \in \mathbb{R}^K$, in which higher score denotes higher ranking. If we have the oracle ranking $r_K^*$, then if $m$-th candidate is favored than $n$-th candidate we can get $r_m^* - r_n^* > 0$. As a result, we can formulate the ranking problem as follows,

$$\min_r \sum_{i=1}^{N_p} \sum_{(m,n) \in E^i} (r_m - r_n - S_{mn}^i)^2. \tag{2}$$

To get the solution we can reformulate Eq. 2 as a Least Square problem by introducing a transformation matrix $D^i \in \mathbb{R}^{\lceil \frac{K}{k} \rceil \times K}$, each row of which is a sparse vector with only the two indices from the pairwise set $E^i$ as 1 and -1. Then we have the following formulation,

$$\min_r \sum_{i=1}^{N_p} \frac{1}{2N_p} \|D^i r - S^i\|_2^2. \tag{3}$$

By solving Eq. 3 we can directly get a proper global ranking $r$. Furthermore, multiple ranking models can be engaged in this process by extending $\mathcal{S}$ with preference matrices calculated from these different models, which can in turn enhance the ranking process with more comprehensive information. Compared with the naive ranking method, this method involves more partial observations from $\mathcal{X}_R$, while avoiding the accumulative error, thus being both effective and efficient.

**Analysis.** It is straightforward to see the merits of the proposed method. First, compared with the naive strategy, solving Eq. 3 can directly lead to a complete ranking results, avoiding repetitive computation for more top ranked examples. Second, it is inevitable that the trained ranking model

wrongly predicts the ranking of some alternatives. Under such circumstance, the naive strategy has now way to fix such an error. On the contrary, since a same alternative pair can result in different predictions by our list-wise ranker, the proposed algorithm as in Alg. 1 can take into account all contradicting predictions, thus leaving potential for rectifying the wrong predictions.

# 4 Experiments

## 4.1 Dataset and setting

**Dataset.** We follow MAE-VQGAN [1] and VPR [32] to adopt three tasks including foreground segmentation, single object detection and colorization. For foreground segmentation, Pascal-$5^i$ [19] is utilized which contains 4 data splits. We conduct experiments and report the mean intersection over union (mIoU) on all splits together with the averaged mIoU among these splits. For single object detection, Pascal VOC 2012 [5] is used. Images and predictions are processed the same as in MAE-VQGAN, in which mIoU is adopted as metric. For colorization, we first sample 50000 training data from ILSVRC2012 [18] training set. Then a test set randomly sampled from the validation set of ILSVRC2012 is used to test the model with mean squared error (MSE) as metric.

**Implementation Detail.** Considering the training data size for each task, we adopt different sequence length for ranking. Specifically, we train rank-5 and rank-10 models for foreground segmentation and single object detection, while rank-3 and rank-5 models are trained for colorization. The training data is built by first selecting 50 most similar images from the whole training set for each image as its alternative set. Then we randomly sample 20 sequences with the required length from this alternative set for training. For all experiments we adopt DINO v2 [14] as feature extracter. AdamW optimizer [12] is used with learning rate set as 5e-5 and batch size set as 64. We utilize 4 V100 gpus to cover all experiments.

**Competitors.** We choose three methods as our competitors, including MAE-VQGAN, VPR which contains UnsupPR and SupPR and prompt-SelF [21], which mainly facilitates an ensemble strategy. To fairly compare our methods with these previous ones, we report main results of our models both with and without the test-time ensemble.

## 4.2 Main results

The experiment results are shown in Tab. 1. Note that we omit the result of our model with voting strategy for colorization since prompt-SelF does not report this term and there is no fair competitors in this setting.

As in Tab. 1, for all three tasks, our model receives best performance on both variant. Specifically, when not using voting strategy, our model outperforms the strongest competitors SupPR by 2.84 in terms of average segmentation mIoU, while the superiority is consistent on other two tasks. When using voting strategy, our model receives about 4 higher mIoU for segmentation and 2 higher mIoU for detection, leading to better results than prompt-SelF. One would note that while SupPR was designed to learn a better metric than the naive visual similarity used for UnsupPR, its performance on colorization is exactly the same as UnsupPR. On contrary to that, our model makes a huge leap with 0.04 less MSE, which is not only much better than both UnsupPR and SupPR but also doubles the improvement these two methods have against the random strategy in MAE-VQGAN. This can prove the effectiveness of our proposed ranking model for selecting in-context examples.

To further illustrate the advantage of Partial2Global, we visualize several samples for segmentation and detection in Fig. 3 and Fig. 5. We find that in some cases the in-context examples selected by VPR can let MAE-VQGAN generate totally wrong results, especially when multiple objects are presented in the query. For example, in the first sample of Fig. 3, the query image contains two monitors, while the target label is only related to the left one. While both VPR and our method select images of monitors as in-context example, VPR results in 0 IoU while prediction using our example is much better. This may be resulted from the spatial relation of the object of interest in the example images. In the example selected by VPR, the monitor is placed at the right, which is the same as the non-target monitor in the query image, thus leading to wrong guidance and prediction. Apart from that, VPR tends to select small objects for detection, as shown in Fig. 5. This can lead to failed detection even though the query objects are large enough for huamns to detect them. In comparison, our method can generally select more proper in-context examples, thus enjoying better performance.

Table 1: Comparison of our method with previous in-context learning methods.

| Model | Seg. (mIoU) ↑ | | | | | Det. (mIoU) ↑ | Color. (MSE) ↓ |
| | Fold-0 | Fold-1 | Fold-2 | Fold-3 | Avg. | | |
|---|---|---|---|---|---|---|---|
| MAE-VQGAN | 28.66 | 30.21 | 27.81 | 23.55 | 27.56 | 25.45 | 0.67 |
| UnsupPR | 34.75 | 35.92 | 32.41 | 31.16 | 33.56 | 26.84 | 0.63 |
| SupPR | 37.08 | 38.43 | 34.40 | 32.32 | 35.56 | 28.22 | 0.63 |
| Ours | **38.81** | **41.54** | **37.25** | **36.01** | **38.40** | **30.66** | **0.58** |
| prompt-SelF | 42.48 | 43.34 | 39.76 | 38.50 | 41.02 | 29.83 | — |
| Ours+voting | **43.23** | **45.50** | **41.79** | **40.22** | **42.69** | **32.52** | — |

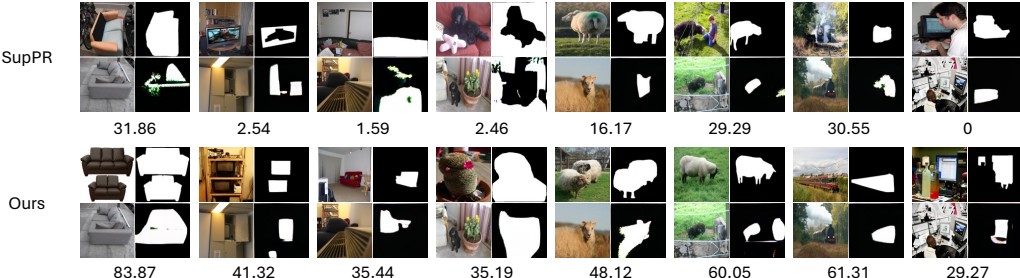

Figure 3: Qualitative comparison between our method and VPR, specifically SupPR, in foreground segmentation. In each item we present the image grid in the same order as the input of MAE-VQGAN, i.e. in-context example and its label in the first row, query image and its prediction in the second row. The IoU is listed below each image grid.

## 4.3 Model analysis

To fully validate the effectiveness of each module in our method, we conduct a series of ablation studies. For all ablation studies, we do not use voting strategy. The experiments contain both foreground segmentation and single object detection if not specified.

**Effectiveness of different modules.** First of all we provide comparison among variants with different designs. Specifically, we consider three models: (1) Rank-10 Naive: A rank-10 model is trained, of which the ranking predictions are directly used for example selection. (2) Rank-10 Aggr.: After training the rank-10 model and getting the ranking predictions, we use the proposed consistency-aware ranking aggregator described in Sec. 3.2 to process these data, then we pick the top-ranked sample in the post-processed global ranking as in-context example. (3) Rank-{5,10} Aggr.: Our full model, in which the ranking prediction of both ranking models are mixed and processed with consistency-aware ranking aggregator to get the final global ranking. The results are presented in Tab. 2. We find that the naive ranking model enjoys 1.54 higher average mIoU than SupPR, which supports our motivation of using ranking model instead of pair-wise contrastive learning model. Since ranking model can make better use of the training data, and comparing multiple samples enables the model to discover the inner relationship between them, rather than simply judging if one sample is better than another one. Moreover, We find that using consistency-aware ranking aggregator to process the ranking predictions can lead to further improvement, which can be enlarged by including results from rank-5 model. This is because the predictions of ranking models can be thought of as from different annotators. Even for a single model, its predictions will not be consistent when given two sequences which share partial same samples. Therefore, simply utilizing the inconsistent predictions can lead to suboptimal results, and consistency-aware ranking aggregator helps regulate the predictions to produce more globally consistent ranking results, thus having better results.

**Is quality of in-context examples really aligned with visual similarity?** A basic claim in VPR is that the more similar an image is to the query image, the better it is as an in-context example. Nonetheless, VPR utilizes performance metrics instead of visual similarity for contrastive learning and provides better results, which makes one wonder if such a claim is indeed solid. To check this

Table 2: Ablation study among different variants of our method.

| Model | Seg. (mIoU) ↑ | | | | | Det. (mIoU) ↑ |
|---|---|---|---|---|---|---|
| | Fold-0 | Fold-1 | Fold-2 | Fold-3 | Avg. | |
| SupPR | 37.08 | 38.43 | 34.40 | 32.32 | 35.56 | 28.22 |
| Rank-10 Naive | 37.51 | 39.69 | 36.62 | 34.58 | 37.10 | 29.58 |
| Rank-10 Aggr. | 38.70 | 41.08 | 37.04 | 35.54 | 38.09 | 29.79 |
| Rank-{5,10} Aggr. | **38.81** | **41.54** | **37.25** | **36.01** | **38.40** | **30.66** |

out, we take a look at the best performing in-context examples selected by our method. The results are shown in Fig. 4(a) and (b). We first visualize the correlation between visual similarity, which is computed by extracting CLIP features for each image and then calculating and averaging the cosine similarity between the alternatives and query, and mIoU on segmentation for both VPR and our method. It is clear that visual similarity can be a basic heuristic for choosing visual in-context examples. Generally well performing examples appear to share high similarity with query samples. On the other hand, there are also large amount of failure cases with high similarity, which indicates the ground principle should be much more complex than visual similarity. Based on this conclusion, we further visualize several cases in Fig. 4(c) selected by our method, which better shows that one should consider more than visual similarity and stress other factors such as object size, spatial position. On the opposite, we find that the background similarity, which can contribute significantly to the visual similarity, is hardly related to the quality of in-context examples.

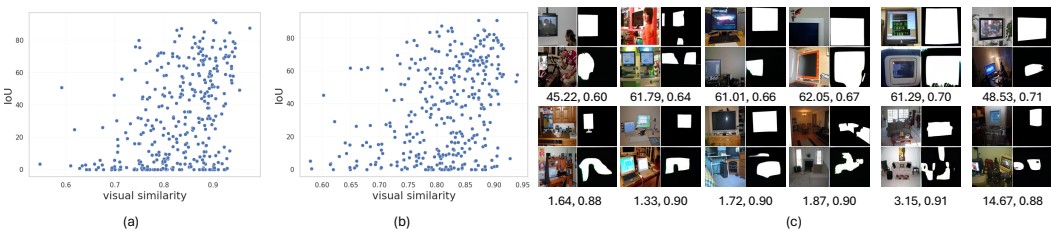

Figure 4: (a) Scatter plot of visual similarity against IoU for VPR on segmentation. (b) Scatter plot of visual similarity against IoU for our method on segmentation. (c) Visualization of several cases with uncorrelated visual similarity and IoU. The first row presents samples with low similarity but proper in-context performance. The second row presents samples with high similarity but poor in-context performance. Captions below each image grid denote IoU and visual similarity sequentially.

**Robustness among different backbones.** We compare our model with two variants with other pretrained feature extractor: CLIP ViT-L/14 and DINO v1. The results are shown in Tab. 3. First, directly using ranking prediction from our ranking models through the naive ranking strategy performs better than VPR no matter which backbone is utilized. VPR also conducted such an experiment to try their method with different backbones such as CLIP, EVA and supervised ViT, but none of those results can outperform ours, which illustrates the efficacy of our design of ranking model against the contrastive learning based model. Second, adopting consistency-aware ranking aggregator consistently improves the naive ranking method by nearly 1-2 mIoU for both segmentation and detection, which is compatible with the results in Tab. 2, showing the robustness of the proposed method among different backbone choices. Third, the in-context performance is not totally correlated with the capacity of backbones. It is commonly known that DINO v2, as an improved version of DINO v1 with more comprehensive objective functions and more data, should be equipped with higher capacity. However, we find that DINO v2 performs worse than DINO v1 on segmentation whether naive ranking or consistency-aware ranking aggregator is adopted. We think such results can be attributed to the different learning difficulty when using these backbones, and it would be interesting to have further research on the impact of backbones for visual in-context learning.

**Consistent optimality of ranking results.** Apart from the strong performance of our method, we further find that the proposed consistency-aware ranking aggregator can interestingly introduce consistent optimality to the ranking results. That is to say, apart from the top ranked sample, the second best and third best samples should also be generally better than other samples in the alternative

Table 3: Ablation study among different backbones of our method.

| Backbone | Strategy | Seg. (mIoU) ↑ | | | | | Det. (mIoU) ↑ |
| | | Fold-0 | Fold-1 | Fold-2 | Fold-3 | Avg. | |
|---|---|---|---|---|---|---|---|
| CLIP | Naive | 37.37 | 40.11 | 36.84 | 33.88 | 37.05 | 29.69 |
| | Aggr. | 38.58 | 41.34 | 37.66 | 35.91 | 38.37 | 30.79 |
| DINOv1 | Naive | 38.78 | 40.02 | 36.92 | 35.12 | 37.71 | 28.03 |
| | Aggr. | 39.25 | 42.27 | 38.45 | 36.77 | 39.19 | 29.19 |
| DINOv2 | Naive | 37.51 | 39.69 | 36.62 | 34.58 | 37.10 | 29.58 |
| | Aggr. | 38.81 | 41.54 | 37.25 | 36.01 | 38.40 | 30.66 |

Table 4: Ablation study among different example selection strategies of our method.

| Strategy | Seg. (mIoU) ↑ | | | | | Det. (mIoU) ↑ |
| | Fold-0 | Fold-1 | Fold-2 | Fold-3 | Avg. | |
|---|---|---|---|---|---|---|
| #1 rank | 38.81 | 41.54 | 37.25 | 36.01 | 38.40 | 30.66 |
| #2 rank | 38.13 | 41.66 | 37.62 | 35.35 | 38.19 | 30.76 |
| #3 rank | 38.66 | 41.08 | 37.36 | 35.91 | 38.25 | 30.61 |
| top2 fusion | 39.08 | 42.61 | 38.17 | 36.67 | 39.13 | 30.16 |
| top3 fusion | 40.07 | 42.48 | 38.77 | 37.61 | 39.73 | 31.85 |
| top5 fusion | 40.12 | 42.59 | 39.09 | 37.28 | 39.77 | 32.08 |

set as in-context example. To show that we test two strategies: (1) directly using samples ranked second and third as in-context examples and (2) adopting a simple late fusion method to average predictions generated with 2, 3, 5 top ranked samples. The results are shown in Tab. 4. One can find that using top-3 best ranked sample have comparable results. For segmentation, the #2 rank samples only have 0.21 lower mIoU than #1 rank samples, and #3 rank samples are even slightly better than #2 rank samples, which evidences the consistent optimality. The results are consistent for detection. Moreover, fusing 2 or 3 samples together leads to 0.73 and 1.43 higher average mIoU, which also supports the consistent optimality brought by the consistency-aware ranking aggregator. Such improvement saturates when using 5 samples, indicating that samples ranked 5-th or later are significantly worse and cannot provide more useful information for the in-context inference.

## 5   Conclusion and discussion

This paper proposes a novel pipeline for in-context example selection in Visual In-Context Learning (VICL). Specifically, we design a transformer-based ranking model, which can provide desirable ranking predictions by well utilizing the inner relationship among alternatives. The ranking results are further aggregated by the proposed consistent-aware ranking aggregator, which achieves desirable global ranking prediction by transforming the problem into a least square problem. Our method receives state-of-the-art performance on three different tasks including foreground segmentation, single object detection and image colorization, showing great potential of improving VICL researches.

**Limitations.** While our proposed method can provide good in-context examples, the basic performance still highly relies on the quality of the in-context learning model. The MAE-VQGAN used in this paper is rather an early stage trial for visual in-context learning, with many limitations in terms of its structures and applications. We believe that our model can further help other stronger visual in-context learning methods, leading to better in-context performance.

**Broader impacts.** Our work will not lead to significant negative social impacts. One main problem is that if the data used for in-context learning contains biases, the models trained on this data may also exhibit these biases, leading to unfair or discriminatory outcomes. This is particularly concerning in areas like facial recognition or predictive policing, where biased models could disproportionately affect certain groups. Solving such problems would be a large future research topic.

## Acknowledgments and Disclosure of Funding

Y. Yao was in part supported by the HKRGC GRF-16308321 and the NSFC/RGC Joint Research Scheme Grant N_HKUST635/20. The computations in this research were performed using the CFFF platform of Fudan University. This project is supported by AI for Science Foundation of Fudan University.

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

# A   More details

**NeuralNDCG.**   The list-wise ranking loss NeuralNDCG adopted in this paper is a differentiable approximation of Normalised Discounted Cumulative Gain (NDCG). We simply quote the definition here from [16] for better understanding. Specifically, for $x$ denoting a sample and $y$ denoting its score, NDCG can be calculated as

$$DCG(\pi, y) = \sum_{i=1}^{n} \frac{2^{y_i} - 1}{\log_2(1 + \pi(i))} \tag{4}$$

$$NDCG(\pi_f, y) = \frac{DCG(\pi_f, y)}{DCG(\pi^*, y)} \tag{5}$$

where $\pi_f$ denotes the predicted ranking, $\pi*$ is the ground truth ranking regarding score $y$. Neural-NDCG works by substituting the discontinuous sorting operator with NeuralSort, which results in an approximated permutation matrix

$$\hat{P}_{sort}(s)\,[i, :]\,(\tau) = softmax\left[\frac{(n + 1 - 2i)s - A_s \mathbb{1}}{\tau}\right] \tag{6}$$

where $A_x[i, j] = |s_i - s_j|$, $\mathbb{1}$ denotes a vector filled with value 1, $\tau$ is a temperature parameter. Then the NeuralNDCG can be calculated as

$$\hat{DCG}(\tau)(\pi, y) = \sum_{i=1}^{n} \frac{scale(\hat{P}2^{y_i} - 1)}{\log_2(1 + \pi(i))} \tag{7}$$

$$NeuralNDCG(\tau)(\pi_f, y) = \frac{\hat{DCG}(\tau)(\pi, y)}{DCG(\pi^*, y)} \tag{8}$$

where $scale(\cdot)$ is Sinkhorn scaling.

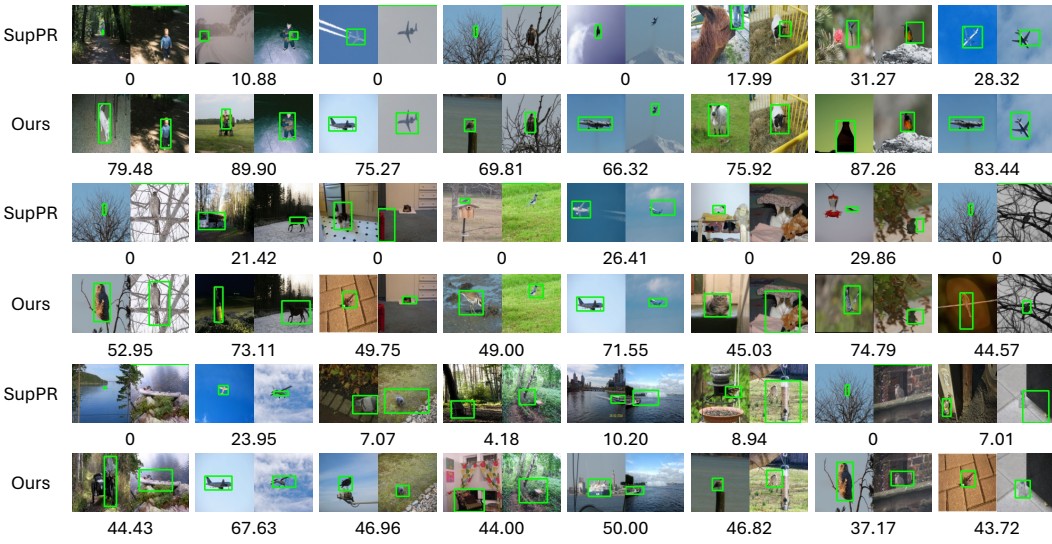

Figure 5: Qualitative comparison between our method and VPR, specifically SupPR, in single object detection. For simplicity we present the bounding boxes on images instead of showing the image grids. In each item the left image denotes the in-context example and the right one denotes the query.

# B   Additional ablation study

**Transferability of Partial2Global.** To test the transferability of the prompt selection method and further reveal the potential of our method, we can add the demonstration suggested by conducting

the following experiment for both SupPR and our method: we use models trained on each fold of segmentation and apply it to other folds. The results are shown in Tab. 6. We find that while both two methods degrade in the transfer learning setting, our method still outperforms SupPR in general. This interesting results indicate that, the training data size of each fold is insuifficient for training a robust and generalizable ranking model. On the other hand, this again indicates that prompt selection for VICL cannot be simply based on visual similarity, as claimed in our paper.

Table 5: Cross-fold performance of our method on segmentation task.

| Source/Target Fold | 0 | 1 | 2 | 3 |
|---|---|---|---|---|
| 0 | — | 36.38 | 32.63 | 30.90 |
| 1 | 35.74 | — | 32.94 | 31.32 |
| 2 | 34.16 | 36.16 | — | 30.44 |
| 3 | 34.28 | 35.93 | 32.98 | — |

Table 6: Cross-fold performance of SupPR on segmentation task.

| Source/Target Fold | 0 | 1 | 2 | 3 |
|---|---|---|---|---|
| 0 | — | 35.46 | 32.44 | 30.95 |
| 1 | 34.92 | — | 32.96 | 31.03 |
| 2 | 34.71 | 36.48 | — | 30.08 |
| 3 | 34.01 | 35.83 | 32.15 | — |

**Efficiency of our method.** One would ask if the proposed method would suffer from poor efficiency, compared with SupPR. In general, the usage of list-wise ranker and the ranking aggregation will inevitably introduce additional computational cost, while the increased complexity during inference is affordable under common circumstances. Specifically, we provide the training and inference time cost as follows. (1) The training of list-wise ranker on the colorization task, which contains about 500000 ranking sequences, takes about 10 hours on 8 V100s. Once the model is trained it can be directly used for any other queries with the same ranking criteria as the training task without any further finetuning. (2) During inference on one V100 gpu, our proposed pipeline requires about 1.17s to rank 50 alternatives for each query sample in the complete process, including feature extracting (0.3s), sub-sequence ranking with list-wise ranker (0.8s), and ranking aggregation (0.07s). Note that some techniques could be utilized to accelerate this process. For example, when we prepare the extracted features in advance (which is reasonable given the candidate set can be prepared in advance), we can skip the feature extraction stage and reduce the time cost by 0.3s. With engineering works, the inference time cost can be further reduced. The detailed inference speed (feature extraction included) given different alternative set size is presented as in Tab. 7.

Table 7: Inference speed with different alternative set size.

| alternative set size | inference time for each query (s) |
|---|---|
| 25 | 1.03 |
| 50 | 1.17 |
| 100 | 1.40 |

**Upper bound of selecting different in-context prompts.** Another interesting problem is to examine the 'upper bound' of performance by adopting different in-context prompts. To further provide insight to this task, we try to examine an upper bound: directly testing all alternatives for each query in the segmentation task and presenting the best IoU in Tab. 8. While our proposed method is much better than SupPR, it still leaves great potential for better performance, which we will take as future works.

Table 8: Oracle in-context learning performance on segmentation.

| | fold0 | fold1 | fold2 | fold3 |
|---|---|---|---|---|
| SupPR | 37.08 | 38.43 | 34.40 | 32.32 |
| Ours | 38.81 | 41.54 | 37.25 | 36.01 |
| best iou among 50 alternative (oracle) | 48.75 | 52.62 | 49.75 | 49.03 |

**Sensitivity test for hyper-parameters.** In general, our method is robust against changes of hyper-parameters. To show this, we try the suggestions to conduct ablation studies on these two hyper-parameters, whose results are presented in the following table. For delta which denotes the margin, using 0 margin leads to worse results while larger margins would be better for learning ranking models. For tau which is the temperature coefficient in NeuralNDCG, we simply use the best setting tau=1 from the original paper. As can be seen in Tab. 9, using smaller temperature would not lead to better results. Nonetheless, all hyperparameter settings enjoy better performance than SupPR, validating the effectiveness of our method.

Table 9: Ablation study for hyper-parameters.

|  | $\delta = 0$ | $\delta = 2$ | $\tau = 0.01$ | $\tau = 0.1$ |
|---|---|---|---|---|
| MSE | 0.594 | 0.588 | 0.608 | 0.592 |

**Impact of alternative set size on performance.** Our choice of alternative set is the same as VPR, i.e. selecting 50 most visually similar samples based on CLIP features. Typically, this choice is good enough as shown in previous works like VPR. Here, to further investigate the impact of data quality, we thus tried the suggestion to test our method on all folds of segmentation task with 25 or 100 alternatives for each query. The results are shown in Tab. 10. One can find that our method is also very robust against different sizes of alternative set.

Table 10: Ablation study for different alternative set sizes.

| set size | fold0 | fold1 | fold2 | fold3 |
|---|---|---|---|---|
| 25 | 38.48 | 41.82 | 37.14 | 35.60 |
| 50 (main) | 38.81 | 41.54 | 37.25 | 36.01 |
| 100 | 38.81 | 41.80 | 37.90 | 36.00 |

**Effectiveness of different terms in our proposed loss.** We have conducted an additional ablation study to compare models trained for colorization without each loss term. The results are shown in Tab. 11. In general all three loss terms contribute the final performance, with $\mathcal{L}_{sort}$ plays the most important role.

Table 11: Ablation study for loss terms.

|  | w/o $\mathcal{L}_{sort}$ | w/o $\mathcal{L}_{margin}$ | w/o $\mathcal{L}_{mse}$ | full loss |
|---|---|---|---|---|
| MSE | 0.601 | 0.595 | 0.585 | 0.583 |

