# OpenReview forum: "Towards Global Optimal Visual In-Context Learning Prompt Selection"
_NeurIPS.cc/2024/Conference — NeurIPS 2024 poster_

### Official Review · Reviewer_dbja · 2024-07-10

**Soundness:** 2
**Presentation:** 2
**Contribution:** 2
**Rating:** 5
**Confidence:** 5

**Summary:**

The paper proposes a framework called Partial2Global for ranking in-context examples in visual language models. This method uses a transformer-based list-wise ranker and a consistency aware ranking aggregator to approximate the global optimal prompt selection. The framework is validated through experiments on tasks such as foreground segmentation, single object detection, and image colorization, demonstrating its effectiveness over existing methods.

**Strengths:**

1：In the analysis, a relatively comprehensive examination was conducted on the sensitivity of different components and various backbones to the order of ICL selection, demonstrating the importance of reasonable prompt ranking for ICL in segmentation tasks.
2：The writing and expression are relatively good.
3：The reasons why sample ranking in ICL may affect the final outcome are analyzed in this paper.

**Weaknesses:**

1.The workload is relatively low. I think this work should also explore the impact of models with a larger number of shots on ICL. Additionally, it should investigate the impact of data quality in the alternative dataset, and include the importance of different parts of the loss in the ranking model during ablation experiments in formula (1).
2.The conclusions presented in some charts are not intuitive. For example, in Figure 4, can the similarity of our method and the visual similarity of VPR be displayed in the same chart?
3.Compared to previous methods, although it achieves a much better level, the improvement is relatively small.

**Questions:**

See weakness part

---

> ### Author Rebuttal · Authors · 2024-08-06
>
> **Q1: The workload is relatively low. I think this work should also explore the impact of models with a larger number of shots on ICL. Additionally, it should investigate the impact of data quality in the alternative dataset, and include the importance of different parts of the loss in the ranking model during ablation experiments in formula (1).**
>
> A1: (1) Larger number shots:Thanks, yes, we had tried many number of shots in our experiments, which are solid enough to support our contributions in generally. And we further add more number of shots in rebuttal. Particularly, our choice of number of shot can be referred to previous methods such as MAE-VQGAN and VPR. As for multi-shot experiment, we in this paper present two trials of multi-shot experiments. First, in Tab.1, we adopt a voting strategy to merge predictions of different permutations of each alternative-query pair. Second, in Tab.4, we present multi-alternative fusion. Both results indicate the efficacy of our method. To further verify the role of multi-alternative fusion, we extend experiments in Tab.4 to top-7 and top-10 fusion. The results are shown in the following table. We can find that the results are consistent with the original conclusion, where the performance is nearly saturated with 5 alternatives. This again supports that the experiments in our paper well-supports our claim.
>
> | \#shot | fold0 | fold1 | fold2 | fold3 | Avg |
> | :---- | :---- | :---- | :---- | :---- | :---- |
> | top2 fusion | 39.08 | 42.61 | 38.17 | 36.67 | 39.13 |
> | top3 fusion | 40.07 | 42.48 | 38.77 | 37.61 | 39.73 |
> | top5 fusion | 40.12 | 42.59 | 39.09 | 37.28 | 39.77 |
> | top7 fusion | 40.14 | 42.36 | 39.01 | 37.32 | 39.71 |
> | top10 fusion | 40.14 | 42.43 | 39.09 | 36.66 | 39.58 |
>
> (2) Impact of data quality. Our choice of alternative set is the same as VPR, i.e. selecting 50 most visually similar samples based on CLIP features. Typically, this choice is good enough as shown in previous works like VPR. Here, to further investigate the impact of data quality, we thus tried the suggestion to test our method on all folds of segmentation task with 25 or 100 alternatives for each query. The results are shown in the following table. One can find that our method is also very robust against different sizes of alternative set.
>
> alternative set size
>
> | set size | fold0 | fold1 | fold2 | fold3 |
> | :---- | :---- | :---- | :---- | :---- |
> | 25 | 38.48 | 41.82 | 37.14 | 35.60 |
> | 50(main) | 38.81 | 41.54 | 37.25 | 36.01 |
> | 100 | 38.81 | 41.80 | 37.90 | 36.00 |
>
> (3) Different parts of the loss. Thanks. We have conducted an additional ablation study to compare models trained for colorization without each loss term. The results are shown in the following table. In general all three loss terms contribute the final performance, with L\_sort plays the most important role.
>
> Loss terms
>
> | colorization | MSE |
> | :---- | :---- |
> | w/o L\_sort | 0.601 |
> | w/o L\_margin | 0.595 |
> | w/o L\_mse | 0.585 |
> | full loss | 0.583 |
>
> **Q2: The conclusions presented in some charts are not intuitive. For example, in Figure 4, can the similarity of our method and the visual similarity of VPR be displayed in the same chart?**
>
> A2: Thank you, but as analyzed in L310-L314, visualization in Fig.4(a,b) is used to show that visual similarity can only act as basic heuristic for in-context example selection. Since we do not intend to compare results in these two figures, presenting results for two methods respectively is enough to support our claim. We will clarify this in our paper.
>
> **Q3: Compared to previous methods, although it achieves a much better level, the improvement is relatively small.**
>
> A3: For better understanding we summarize the improvement of both our method and SupPR against UnsupPR in the following table. Concretely, for segmentation and detection improvement of our method against SupPR is comparable to that of SupPR against UnsupPR. For colorization SupPR cannot improve based on UnsupPR while our method improves it by 0.05 MSE. These results are significant enough to validate the effectiveness of our proposed method.
>
> | Model | Seg-fold0 | Seg-fold1 | Seg-fold2 | Seg-fold3 | Det. | Color. |
> | :---- | :---- | :---- | :---- | :---- | :---- | :---- |
> | SupPR | \+2.33 | \+2.51 | \+1.99 | \+1.16 | \+1.38 | \-0.00 |
> | Ours | \+4.06 | \+5.62 | \+4.84 | \+4.85 | \+3.82 | \-0.05 |

---

> > ### Comment · Reviewer_dbja · 2024-08-12
> >
> > Most of my concerns are solved, Thanks for the response.

---

### Official Review · Reviewer_PwDC · 2024-07-11

**Soundness:** 3
**Presentation:** 3
**Contribution:** 3
**Rating:** 6
**Confidence:** 3

**Summary:**

This paper addresses the fundamental problem in Visual In-Context Learning (VICL), which is how to select the best prompt. Specifically, it focuses on the ranking problem. The paper proposes an algorithm called Partial2Global, which conducts an in-context example selection framework to find the global optimal prompt. The effectiveness of the proposed algorithm is demonstrated through several tasks, such as segmentation, object detection, and colorization.

**Strengths:**

(1) The algorithm is simple yet intuitive for finding the optimal global ranking using a list-wise ranker. It adopts a consistency-aware ranking aggregator.

(2) The performance is evaluated across various tasks, showing improved results in those tasks.

**Weaknesses:**

The method seems to incur additional costs due to the ranking process. Is there any further analysis on this?

**Questions:**

Please see the weakness part.

**Limitations:**

Please see the weakness part.

---

> ### Author Rebuttal · Authors · 2024-08-06
>
> **Q1: The method seems to incur additional costs due to the ranking process. Is there any further analysis on this?**
>
> A1: Thank you. Please refer to the general response. Generally, we can summarize that the usage of list-wise ranker and the ranking aggregation will inevitably introduce additional computational cost, while the increased complexity during inference is affordable in our experiments.

---

### Official Review · Reviewer_dHhp · 2024-07-12

**Soundness:** 3
**Presentation:** 3
**Contribution:** 3
**Rating:** 5
**Confidence:** 4

**Summary:**

This paper addresses Visual In-Context Learning (VICL), which uses examples to help models learn new tasks. The main challenge is choosing the best prompt to improve learning and prediction. The authors introduce Partial2Global, a new method to find the best examples for each query. This method uses a transformer-based ranker to compare alternatives and a ranking aggregator for consistency. Experiments show that Partial2Global outperforms other methods in tasks like segmentation, object detection, and colorization, setting new performance standards.

**Strengths:**

1.The proposed method, Partial2Global, introduces a novel framework for in-context example selection that addresses the fundamental challenge in Visual In-Context Learning (VICL) with a unique transformer-based list-wise ranker and a consistency-aware ranking aggregator.

2.The authors have conducted extensive experiments across multiple tasks, including foreground segmentation, single object detection, and image colorization. This robust evaluation provides strong evidence of the method's effectiveness and generalizability.

3.  The paper is well-written and easy to follow, with complex ideas clearly explained. In particular, Figure 2 offers a clear and detailed comparison of existing methods, helping to contextualize the contributions of the proposed approach.

**Weaknesses:**

1.The paper does not include ablation studies on the critical parameters, such as δ and τ. These studies are essential for understanding how variations in these parameters affect the model's performance. Without this information, it is difficult to assess the robustness and sensitivity of the model to changes in these parameters.

2.The paper introduces heuristic methods for ranking in-context examples using a list-wise ranker and a consistency-aware ranking aggregator. However, there is no analysis provided on the computational complexity and time cost associated with these methods. Given the potential for high computational demands, especially with large datasets, it is important to understand the scalability and practicality of these approaches.

3.Figure 4 of the paper presents results related to visual similarity, but it lacks a clear explanation of how this similarity is computed. The method or formula used to calculate visual similarity is not provided, which is crucial for the reproducibility of the results and for understanding the underlying methodology.

**Questions:**

Please see Weaknesses.

**Limitations:**

The authors address the limitations and potential negative societal impacts of their work.

---

> ### Author Rebuttal · Authors · 2024-08-06
>
> **Q1: The paper does not include ablation studies on the critical parameters, such as δ and τ (NDCG). These studies are essential for understanding how variations in these parameters affect the model's performance. Without this information, it is difficult to assess the robustness and sensitivity of the model to changes in these parameters.**
>
> A1: In general, our method is robust against changes of hyper-parameters. To show this, we  try the suggestions to conduct ablation studies on these two hyper-parameters, whose results are presented in the following table. For delta which denotes the margin, using 0 margin leads to worse results while larger margins would be better for learning ranking models. For tau which is the temperature coefficient in NeuralNDCG, we simply use the best setting tau=1 from the original paper. As can be seen in the following table, using smaller temperature would not lead to better results. Nonetheless, all hyperparameter settings enjoy better performance than SupPR, validating the effectiveness of our method.
>
> |  | MSE |
> | :---- | :---- |
> | $\delta$=0 | 0.594 |
> | $\delta$=2 | 0.588 |
> | $\tau$=0.01 | 0.608 |
> | $\tau$=0.1 | 0.592 |
>
> **Q2: The paper introduces heuristic methods for ranking in-context examples using a list-wise ranker and a consistency-aware ranking aggregator. However, there is no analysis provided on the computational complexity and time cost associated with these methods. Given the potential for high computational demands, especially with large datasets, it is important to understand the scalability and practicality of these approaches.**
>
> A2: Thank you. Please refer to the general response. We give a thorough analysis. Generally, we can summarize that the usage of list-wise ranker and the ranking aggregation will inevitably introduce additional computational cost, while the increased complexity during inference is affordable in our experiments.
>
> **Q3: Figure 4 of the paper presents results related to visual similarity, but it lacks a clear explanation of how this similarity is computed. The method or formula used to calculate visual similarity is not provided, which is crucial for the reproducibility of the results and for understanding the underlying methodology.**
>
> A3: Thank you. As illustrated in L310, we use commonly used CLIP to extract features for each image, and then we calculate the cosine similarity between the alternatives and query, which are then averaged. We will add the details to our paper.

---

> > ### Comment · Reviewer_dHhp · 2024-08-12
> >
> > I appreciate the author's response. I have read the author's responses, and most of my concerns have been addressed. Therefore, I have decided to maintain a 'Borderline Accept' stance on this paper.

---

### Official Review · Reviewer_crFX · 2024-07-29

**Soundness:** 3
**Presentation:** 2
**Contribution:** 3
**Rating:** 6
**Confidence:** 3

**Summary:**

This paper studies the demonstration retrieval mechanism for visual in-context learning. The authors propose Partial2Global which uses a transformer-based ranker and a consistency-aware aggregator to find the optimal demonstration. Experiments show that Partial2Global outperforms existing methods in tasks like segmentation and object detection, setting new state-of-the-art results.

**Strengths:**

1. The paper is well-written and easy to follow.

2. The motivation is clear and reasonable.

3. The proposed method is interesting and novel. Exploring efficient and effective global ranking in place of the previous local information and using least-squares fusion to mitigate iterative loss.

4. The ablation studies are adequate, revealing the effectiveness of global information, the efficacy of the fusion method, and that similar images do not necessarily yield better contextual effects. It also identifies defects in samples retrieved by methods like VPR (e.g., frequently retrieving smaller objects in detection tasks, which results in poorer VICL performance).

**Weaknesses:**

1. Since training exclusively on a single dataset is limiting, I would appreciate a demonstration of transfer learning performance to prove its adaptability and universality across a broad range of data.

2. It would be beneficial to analyze the efficiency of your model to the size of the retrieval set, as complexity is a crucial aspect.

3. It would be insightful to examine the "upper bound" of performance, such as by concatenating each demonstration with a query to see what the maximum Intersection over Union (IoU) is, and to assess how much room there is for improvement from this upper bound.

**Questions:**

See the Weaknesses.

**Limitations:**

N/A.

---

> ### Author Rebuttal · Authors · 2024-08-06
>
> **Q1: Since training exclusively on a single dataset is limiting, I would appreciate a demonstration of transfer learning performance to prove its adaptability and universality across a broad range of data.**
>
> A1: Thank you for the suggestion. We would like to highlight that the current experiment results can sufficiently validate the efficacy of our method. Nevertheless, to test the transferability of the prompt selection method and further reveal the potential of our method, we can add the demonstration suggested by conducting the following experiment for both SupPR and our method: we use models trained on each fold of segmentation and apply it to other folds. The results are shown in the following two tables. We find that while both two methods degrade in the transfer learning setting, our method still outperforms SupPR in general.
> This interesting results indicate that, the training data size of each fold is insuifficient for training a robust and generalizable ranking model. On the other hand, this again indicates that prompt selection for VICL cannot be simply based on visual similarity, as claimed in our paper.
>
> Ours
>
> | source/target fold | 0 | 1 | 2 | 3 |
> | :---- | :---- | :---- | :---- | :---- |
> | 0 | — | 36.38 | 32.63 | 30.90 |
> | 1 | 35.74 | — | 32.94 | 31.32 |
> | 2 | 34.16 | 36.16 | — | 30.44 |
> | 3 | 34.28 | 35.93 | 32.98 | — |
>
> SupPR
>
> | source/target fold | 0 | 1 | 2 | 3 |
> | :---- | :---- | :---- | :---- | :---- |
> | 0 | — | 35.46 | 32.44 | 30.95 |
> | 1 | 34.92 | — | 32.69 | 31.03 |
> | 2 | 34.71 | 36.48 | — | 30.08 |
> | 3 | 34.01 | 35.83 | 32.15 | — |
>
> **Q2: It would be beneficial to analyze the efficiency of your model to the size of the retrieval set, as complexity is a crucial aspect.**
>
> A2: Thank you. Please refer to the general response. Generally, we can summarize that the usage of list-wise ranker and the ranking aggregation will inevitably introduce additional computational cost, while the increased complexity during inference is affordable in our experiments.
>
> **Q3: It would be insightful to examine the "upper bound" of performance, such as by concatenating each demonstration with a query to see what the maximum Intersection over Union (IoU) is, and to assess how much room there is for improvement from this upper bound.**
>
> A3: Thank you for this suggestion. We want to first highlight that our experiment results can already validate the efficacy of our method against the competitor as a powerful in-context prompt selection method. To further provide insight to this task, we try to examine an upper bound: directly testing all alternatives for each query in the segmentation task and presenting the best IoU in the following table. While our proposed method is much better than SupPR, it still leaves great potential for better performance, which we will take as future works.
>
> |  | fold0 | fold1 | fold2 | fold3 |
> | :---- | :---- | :---- | :---- | :---- |
> | SupPR | 37.08 | 38.43 | 34.40 | 32.32 |
> | Ours | 38.81 | 41.54 | 37.25 | 36.01 |
> | best iou among 50 alternative (oracle) | 48.75 | 52.62 | 49.75 | 49.03 |

---

> > ### Comment · Reviewer_crFX · 2024-08-11
> > **Thanks for the response. I maintain my positive score.**
> >
> > I appreciate the authors' response. I have read the response and some of my concerns (Q1 & Q3) have been addressed. I decide to keep a Weak Accept for the paper.

---

### Author Rebuttal · Authors · 2024-08-06

We thank the reviewers for all of your time to write valuable and constructive comments. Your feedback will definitely assist us in enhancing the quality of our paper, and thus we are committed to incorporating these suggestions in our revision process. Meanwhile, we feel encouraged that the reviewers find our paper well-written (Reviewer crFX, dHhp and dbja), our method novel and effective (Reviewer crFX, dHhp, PwDC), and our experiments solid and comprehensive (Reviewer crFX, dHhp and PwDC). Your support means a lot to us\! At this juncture, we would like to illustrate the general concern of training and testing efficiency of our method.

In general, the usage of list-wise ranker and the ranking aggregation will inevitably introduce additional computational cost, while the increased complexity during inference is affordable under common circumstances. Specifically, we provide the training and inference time cost as follows.

1) The training of list-wise ranker on the colorization task, which contains about 500000 ranking sequences, takes about 10 hours on 8 V100s. Once the model is trained it can be directly used for any other queries with the same ranking criteria as the training task without any further finetuning.
2) During inference on one V100 gpu, our proposed pipeline requires about 1.17s to rank 50 alternatives for each query sample in the complete process, including feature extracting (0.3s), sub-sequence ranking with list-wise ranker (0.8s), and ranking aggregation (0.07s). Note that some techniques could be utilized to accelerate this process. For example, when we prepare the extracted features in advance (which is reasonable given the candidate set can be prepared in advance), we can skip the feature extraction stage and reduce the time cost by 0.3s. With engineering works, the inference time cost can be further reduced. The detailed inference speed (feature extraction included) given different alternative set size is presented as follows

| alternative set size | inference time for each query (s) |
| :---- | :---- |
| 25 | 1.03 |
| 50 | 1.17 |
| 100 | 1.40 |

In response to the reviewers' comments, we have thoroughly reviewed our paper, performed additional experiments, and prepared a comprehensive response. We will improve the manuscript according to your comments. We hope that our paper adequately addresses your concerns. We kindly look forward to your recognition.

Best regards.

---

### Decision · Program_Chairs · 2024-09-25

**Decision:**

Accept (poster)

**Comment:**

The paper received two Weak Accept and two Borderline Accept responses. The AC read the paper, rebuttal, and discussions, and agree with the reviewers' recommendations. It is also encouraged that the authors incorporate reviewers' comments in their final version. Congratulations to the acceptance!